# Synthesis, Performance Measurement of Bi_2_SmSbO_7_/ZnBiYO_4_ Heterojunction Photocatalyst and Photocatalytic Degradation of Direct Orange within Dye Wastewater under Visible Light Irradiation

**DOI:** 10.3390/ma15113986

**Published:** 2022-06-03

**Authors:** Jingfei Luan, Bingbing Ma, Ye Yao, Wenlu Liu, Bowen Niu, Guangmin Yang, Zhijie Wei

**Affiliations:** 1School of Physics, Changchun Normal University, Changchun 130032, China; mbb216216@126.com (B.M.); yaoye1109@mails.jlu.edu.cn (Y.Y.); LWL7200@126.com (W.L.); niubw2021@126.com (B.N.); yangguangmin@ccsfu.edu.cn (G.Y.); weizj2021@126.com (Z.W.); 2State Key Laboratory of Pollution Control and Resource Reuse, School of the Environment, Nanjing University, Nanjing 210093, China

**Keywords:** Bi_2_SmSbO_7_, Bi_2_SmSbO_7_/ZnBiYO_4_ heterojunction, direct orange, photocatalytic degradation, visible light irradiation

## Abstract

Originally, the new catalyst Bi_2_SmSbO_7_ was synthesized by the hydrothermal synthesis method or by the solid-phase sintering method at a lofty temperature. A solvothermal method was utilized to prepare a Bi_2_SmSbO_7_/ZnBiYO_4_ heterojunction photocatalyst (BZHP). The crystal structure of Bi_2_SmSbO_7_ belonged to the pyrochlore structure and face-centered cubic crystal system by the space group of *Fd3m*. The cell parameter a was equivalent to 10.835(1) Å (Bi_2_SmSbO_7_). With Bi_2_SmSbO_7_/ZnBiYO_4_ heterojunction (BZH) as the photocatalyst, the removal rate (RR) of direct orange (DO) and the total organic carbon were 99.10% and 96.21% after visible light irradiation of 160 min (VLI-160M). The kinetic constant k toward DO concentration and visible light irradiation time (VLI) with BZH as photocatalyst reached 2.167 min^−1^. The kinetic constant k, which was concerned with total organic carbon, reached 0.047 min^−1^. The kinetic curve that came from DO degradation with BZH as a catalyst under VLI conformed to the second-order reaction kinetics. After VLI-160M, the photocatalytic degradation (PD) removal percentage of DO with BZH as the photocatalyst was 1.200 times, 1.268 times or 3.019 times that with Bi_2_SmSbO_7_ as the photocatalyst, ZnBiYO_4_ as the photocatalyst or with nitrogen-doped titanium dioxide as the photocatalyst. The photocatalytic activity (PA) was as following: BZH > Bi_2_SmSbO_7_ > ZnBiYO_4_ > nitrogen-doped titanium dioxide. After VLI-160M for three cycles of experiments with BZH as the photocatalyst, the RR of DO reached 98.03%, 96.73% and 95.43%, respectively, which meant that BZHP possessed high stability. By using the experiment of adding a trapping agent, the oxidative purifying capability for degradation of direct orange, which was in gradual depressed order, was as following: hydroxyl radical > superoxide anion > holes. Finally, the possible degradation pathway and degradation mechanism of DO were discussed systematically. A new high active heterojunction catalyst BZHP, which could efficiently remove toxic organic pollutants such as DO from dye wastewater after VLI, was obtained. Our research was meant to improve the photocatalytic property of the single photocatalyst.

## 1. Introduction

Due to high chroma, high chemical oxygen demand and complex composition, dye pollutants from the textile and photographic industries are becoming a serious environmental problem [1,2,3,4,5,6]. Direct orange S (C_33_H_22_N_6_Na_2_O_9_S_2_) was one of the most common pollutants found in wastewater [7,8], it was mainly used for dyeing textiles, leather and paper. Among various dyes, direct orange (DO) dye was very hard to degrade, however, DO was frequently used as a standard dyestuff pollutant for evaluating the activity of a photocatalyst under UV-light shining [8,9,10,11,12,13,14]. Therefore, the effective degradation of direct orange was a problem to be solved.

The conventional processing methods that we used to degrade those dye contaminates were bio-degradation, electrochemistry, adsorption and flocculation-precipitation [15,16,17,18,19,20,21,22]. However, because of the shortcomings and limitations of each method, the conventional methods of treating wastewater cannot achieve the maximum degradation effect [23]. Since 1972, photocatalytic reaction was firstly found, the photocatalysis technology had been booming due to the strong market demand [24], and is widely used in sewage treatment [25,26]. Photocatalysts could produce oxidation groups under light irradiation via absorbing sunlight as the energy source, decompose organic pollutants to generate oxidative free radicals [25,26,27,28,29], and finally effectively remove organic pollutants. Therefore, photocatalysis technology was scientifically attractive because of its high efficiency, energy-saving and pollution-free characteristics [28,29].

Metal oxides [30,31,32,33,34,35,36,37,38,39,40,41] and metal sulfides [38,39,40,41,42,43,44,45], such as TiO_2_ and ZnO, were the most common types of semiconductor photocatalysts. However, the long-term development of TiO_2_ in the market will be limited, because of the wide band gap, meaning that TiO_2_ can only absorb UV-light (occupying 5% of the solar energy), and for this reason it cannot make reasonable use of optical energy [46]. A report on the Ni-doped InTaO_4_ (the chemical formula can be summarized as ABO_4_) compound in 2001 showed that the ABO_4_ compound had great potential for photocatalytic preparation of hydrogen under visible light irradiation (VLI) [47]. Fortunately, in recent years, A_2_B_2_O_7_ compounds as photocatalysts [48,49] have also been reported to be able to degrade pollutants in wastewater. In 2011, for the sake of removing the rhodamine B in wastewater, Luan et al. synthesized and used nano-catalysts Y_2_InSbO_7_ and Y_2_GdSbO_7_ for the first time, and studied their structure and photocatalytic properties [48]. Luan et al. prepared Cd_2_BiSbO_7_ and Gd_2_YSbO_7_ photocatalysts and studied the structure and catalytic performance of a single catalyst. Based on their report, these two catalysts achieved complete removal of rhodamine B, indicating that they were very good visible light-responsive catalysts [49]. As we all know, small improvements in the construction configuration of quasi-conductor catalysts might facilitate the disassociation of photo-generated current carriers, thereby improving photocatalytic activities [50,51,52,53,54,55,56]. There are many effective methods [50,51,52,53,54,55,56,57,58,59,60,61,62,63] which could improve the activity of photocatalysts, such as ions doping, the construction of heterojunctions and photosensitization.

Many methods [50,51,52,53,54,55,56,57,58,59,60,61,62,63] have been proven to be effective, such as ion doping methods, the construction of heterojunctions and photosensitization. Among all the methods mentioned above, the design of composite materials was a promising study sphere of photocatalysts. The composite photocatalyst concentrated the role of a single photocatalyst so that the composite system [50,51,52,53,54,55,56,57,58,59,60,61,62,63] had a higher efficiency of light utilization, photocatalytic performance and chemical stability. As we had reported in the previous work [64], Gd_2_YbSbO_7_ acted as a photocatalyst, crystallized in a pyrochlore-type structure, therefore, changing its structure seemed to be a possible method for realizing the improvement of the PA. According to all the analysis results which were listed above, we could assume that Gd^3+^ in Gd_2_YbSbO_7_ was replaced by Bi^3+^, and the replacement of Yb^3+^ by Sm^3+^ might increase the carrier concentration. Consequently, the electrical transportation and photophysical properties showed an obvious change and improvement in the novel Bi_2_SmSbO_7_ compound, which might possess advanced photocatalytic performance. In addition, the construction of heterojunctions has been proven to be an effective way to enhance the photocatalytic efficiency [65,66,67,68,69,70,71,72,73,74,75,76,77,78,79,80,81,82]. Sun et al. realized that the degradation of ciprofloxacin on BiVO_4_-Bi_2_WO_6_ nano-heterojunction photocatalyst was driven by visible light. In their report, the nanometer heterogenous junction photocatalyst (BiVO_4_-Bi_2_WO_6_) exhibited improved photocatalytic degradation (PD) activity for degrading ciprofloxacin under VLI [65]. Yang et al. prepared the g-C_3_N_4_@BiOCl visible light-responsive photocatalyst with a hollow flower-like structure through a self-assembly strategy. Due to the excellent charge separation ability under VLI, the heterojunction photocatalyst degradation of rhodamine B exhibited much higher photocatalytic activity (PA) than g-C_3_N_4_ and BiOCl [66]. Analyzing the above results, it was known that constructing a heterojunction photocatalyst could not only cause an obvious improvement in the reduction-oxidation property of the catalyst [67] but also improve the reactionary activity. Therefore, a Bi_2_SmSbO_7_/ZnBiYO_4_ heterogenous junction photocatalyst was also synthesized, and the performance of this heterojunction photocatalyst in DO degradation was worth looking forward to.

In the article, an X-ray diffractometer (XRD), scan electronic microscope-energetic disperse spectrum (SEM-EDS) and X-ray phoelectron spectrometer (XPS) were utilized for analyzing the structural properties of pure phase ZnBiYO_4_ and single phase Bi_2_SmSbO_7_. The removal rate (RR) of DO under VLI with pure phase Bi_2_SmSbO_7_ as a catalyst, ZnBiYO_4_ as a catalyst, N-doped TiO_2_ (N-dT) as a catalyst or with Bi_2_SmSbO_7_/ZnBiYO_4_ heterojunction (BZH) as a photocatalyst was detected. Because the energy band width of ZnBiYO_4_ was 1.953 eV, which was easily lower than the energy of incident visible light under VLI, it was easy to generate photo-generated electrons and photo-generated holes which were separated efficiently and could not easily be recombined. Therefore, ZnBiYO_4_ was a visible light responsible catalyst with high photocatalytic activity. Meanwhile, the conduction band potential of ZnBiYO_4_ was −0.682 eV, which was more negative than −0.33 eV, therefore, the photo-generated electrons on the conduction band of ZnBiYO_4_ were more likely to interact with dissolved oxygen in water to generate superoxide anion. The superoxide anion had a strong oxidizing effect and could directly oxidize direct orange. The valence band (VB) electric potential of Bi_2_SmSbO_7_ was 2.539 eV, which was more nonnegative than 2.38 eV. Therefore, holes within the VB of Bi_2_SmSbO_7_ might oxidize hydroxyl and water into hydroxyl radicals for degrading DO in water. Hydroxyl radicals had a strong oxidizing effect; therefore, the pollutant DO in water could be directly and efficiently oxidized by hydroxyl radicals. At the same time, the energy band width of Bi_2_SmSbO_7_ was 2.42 eV, thus Bi_2_SmSbO_7_ was also a visible light responsible catalyst with high photocatalytic activity. Both Bi_2_SmSbO_7_ and ZnBiYO_4_ could generate photo-generated electrons and photo-generated holes which were not easily recombined under VLI, and had high photocatalytic activity, thus Bi_2_SmSbO_7_ and ZnBiYO_4_ could build a perfect heterojunction together. In this study, our purpose was to prepare novel heterojunction catalysts which could remove DO from pharmaceutic wastewater under VLI. For the first time, a new type of A_2_B_2_O_7_ compound Bi_2_SmSbO_7_ nano catalyst was successfully synthetized and the Bi_2_SmSbO_7_/ZnBiYO_4_ heterojunction photocatalyst (BZHP) was proposed for removing DO in dye wastewater more efficiently.

## 2. Result and Discussion

### 2.1. XRD Analysis

The structural property of Bi_2_SmSbO_7_ was confirmed by XRD experiment, and the corresponding results, which were obtained by the Materials Studio program based on the Rietveld analysis method, are shown in Figure 1a. According to the refinement results, Bi_2_SmSbO_7_ was a pure phase, simultaneously, the cell parameter *a* was equivalent to 10.835(1) Å. Based on the truing result, the observed and the calculated intensities showed a highly consistent pattern, which proved that Bi_2_SmSbO_7_ was a cubical syngony with a space group of *Fd3m*, and in the refinement model, O atoms were included. Table 1 shows the atomic coordinates and structural parameters of Bi_2_SmSbO_7_. Figure 1b shows the atomic structure of Bi_2_SmSbO_7_. It could be concluded from Figure 1a that Bi_2_SmSbO_7_ crystallized into a pyrochlore-type structure. The full profile structure refinement results of Bi_2_SmSbO_7_ produced unweighted R factors, R_P_ = 22.09%, and the space group was *Fd3m*.

It was obvious that the variation sign of the crystalloid structure of the A_2_B_2_O_7_ compounds was interrelated with the *x* coordinate of the O (1) atom, when the longness of the A-O (1) bonds was equivalent to the longness of the A-O (2) bonds, and the coordinate was 0.375 [83]. Therefore, if the x value was gained, the information about the distortion of the octahedra (MO_6_ (M = Sm^3+^ and Sb^5+^)) could be confirmed [83]. Based on the x value, which was 0.375, the distortion of the MO_6_ octahedra could be confirmed to exist in the crystalloid structure of Bi_2_SmSbO_7_ [83]. For the purpose of preventing recombination of the photoinduced electrons and photoinduced holes, electric charge disjunction was required for PD of direct orange under VLI. According to the reports of Kohno [84] and Nakagawa [85], the localized torsional deformation of the MO_6_ octahedron would have helped in inhibiting the electric charge reconfiguration and above-mentioned important factors would, in the end, have contributed to the enhancement of the PA. Based on their theoretical basis, it was easy to conjecture that the torsional deformation of the MO_6_ octahedron in the crystalline nature of Bi_2_SmSbO_7_ could also be considered to be useful for enhancing the PA. Bi_2_SmSbO_7_ contained a tridimensional network configuration of corner-sharing SmO_6_ and SbO_6_ octahedrons. Each Bi^3+^ ion was connected to two MO_6_ octahedron to form a chain. There were two lengths of Bi-O bonds, three fourths were 2.687 Å (Bi-O (1)) and the rest were 2.273 Å (Bi-O (2)). The six M-O (1) (M = Sm^3+^ and Sb^5+^) bond lengths were 1.978 Å and the M-Bi (M = Sm^3+^ and Sb^5+^) bond lengths were 4.353 Å. The M-O-M (M = Sm^3+^ and Sb^5+^) bond angles were 139.624° in the crystalline nature of Bi_2_SmSbO_7_. The Bi-M-Bi (M = Sm^3+^ and Sb^5+^) bond angles were 131.743° in the crystalline nature of Bi_2_SmSbO_7_. The Bi-M-O (M = Sm^3+^ and Sb^5+^) bond angles were 135.505° in the crystalline nature of Bi_2_SmSbO_7_. Many previous reports showed that the luminescent properties were related to the bond angle, which was on the verge of 180°. The locomotivity of the photogenerated electrons and photogenerated holes was enhanced [83] and the PA was improved because the electrons and holes could easily get to the reaction sites of the catalyst surface [83].

Moreover, the Sb–O–Sb bond angle of Bi_2_SmSbO_7_ was larger, which resulted in an increase in the PA of Bi_2_SmSbO_7_. In accordance with the above analysis, the effect of degrading direct orange under VLI with Bi_2_SmSbO_7_ as the catalyst was mainly due to the crystalline nature and electronic crystalline nature.

Figure 2 shows the X-ray diffraction pattern of ZnBiYO_4_. We have labeled the individual diffractive peaks in Figure 2. The structure of ZnBiYO_4_ was tested by X-ray diffraction technology. We used the Materials Studio program for analyzing the collected data, and further information about the structure was obtained using the Rietveld analysis method. The conclusion could be made that ZnBiYO_4_ was single phase and the structure cell parameter of ZnBiYO_4_ could be equivalent to a = b = 11.176(5) Å and c = 10.014(3) Å. Based on the refinement result, we proved that the crystal formation of ZnBiYO_4_ went with a quadrigonal spinelle crystalline nature and space group *I41/A*. The band gap of ZnBiYO_4_ was equivalent to 1.953 eV.

Figure 3 reveals the XRD spectrum of the Bi_2_SmSbO_7_/ZnBiYO_4_ heterojunction photocatalyst. It could be seen from Figure 3 that the pure single-crystal photocatalyst Bi_2_SmSbO_7_ and single phase photocatalyst ZnBiYO_4_ existed. Each diffraction peak of Bi_2_SmSbO_7_ and each diffraction peak of ZnBiYO_4_ were successfully marked, and other impurities were not found in Figure 3.

### 2.2. Diffused Reflection Spectrum 

The absorbing spectra of the Bi_2_SmSbO_7_ sample are listed in Figure 4a,b. The absorption edge of this new photocatalyst Bi_2_SmSbO_7_ was at 458 nm which could be found within the seeable light range. The bandgap energy (BGE) of the crystal quasi-conductor was calculated by the Kubelka–Munk function (1) (known as the re-emission function) [86,87].
(1)$$\frac{{\left[1-R_{d}\left(hv\right)\right]}^{2}}{2R_{d}\left(hv\right)}=\frac{\alpha \left(hv\right)}{S}$$

In this function, *S* represented the scattered factor, *R_d_* represented the diffuse reflectance, and *α* was absorbance index of radiation.

The absorption which was in the region of the energy band edging of the crystal quasi-conductor followed the equality as (2) [88,89]: (2)$$\alpha hv=A{\left(hv-E_{g}\right)}^{n}$$

In this equation, A represents the proportional constant, α represents absorption coefficient, *E_g_* represents band gap and *ν* represents light frequency, and n determined the transition property of the quasi-conductor. 

Following the above procedures, the values of *E_g_* for Bi_2_SmSbO_7_ could be estimated as 2.42 eV. The estimated merical number of n could be equivalent to about 2, which indirectly allowed the optical transition of Bi_2_SmSbO_7_.

Figure 5a,b show the UV-Vis diffuse reflectance spectra of ZnBiYO_4_. According to the above procedures and Figure 5a,b, the merical number of *E_g_* for ZnBiYO_4_ could be estimated as 1.95 eV. The estimated merical number of n was approximately 2, which indirectly allowed the photic transition of ZnBiYO_4_.

Figure 6a,b show the UV-Vis diffuse reflectance spectra of BZH. In accordance with the above procedures and Figure 6a,b, the numerical value of *E_g_* for BZH was calculated to be 2.15 eV. The estimated merical number of n was equivalent to approximately 2, which indirectly allowed the photic transition of BZH.

According to Formula (2), we obtained the following formula (*αhν*)^1/2^ = *hν* − *E_g_*, and (*αhν*)^1/2^ was regarded as the *y* coordinate and *hν* was regarded as the *x* coordinate. According to Figure 6b, the intersection with the *x* axis should be the value of *E_g_* = 2.15 eV when y = 0. The BGE of Bi_2_SmSbO_7_ was 2.42 eV; the BGE of ZnBiYO_4_ was 1.95 eV; the BGE of BZH was 2.15 eV; the BGE of Bi_3_O_5_I_2_ could be equivalent to 2.02 eV [90]; and the BGE of zinc oxide, which was doped with cobalt, could be equivalent to 2.39 eV [91]. Every BGE of the above five compounds was lower than 2.43 eV, meaning that all the compounds would not only show responsive characteristics under VLI, but also possess tremendous potential to exhibit lofty photocatalytic activity.

### 2.3. Performance Representation of Bi_2_SmSbO_7_/ZnBiYO_4_ Heterojunction Photocatalyst

So as to obtain the valence states and the surface chemical compositions of each element of Bi_2_SmSbO_7_/ZnBiYO_4_, the XPS was accomplished. Figure 7 shows the XPS comprehensive spectrogram of Bi_2_SmSbO_7_/ZnBiYO_4_. Figure 8 represents the XPS spectrogram of O^2−^, Bi^3+^, Sm^3+^, Zn^2+^, Y^3+^ and Sb^5+^, which are derived from Bi_2_SmSbO_7_/ZnBiYO_4_. Based on the XPS full spectrum, which was shown in Figure 7, the synthesized Bi_2_SmSbO_7_/ZnBiYO_4_ included the elements of Bi, Sm, Zn, Y, Sb and O. According to XPS analysis results, which was displayed in Figure 7 and Figure 8, the chemical valence of Bi, Sm, Zn, Bi, Y, Sb or O ion was equivalent to +3, +3, +2, +3, +3, +5 or −2. According to the results of the above analytical tests, the formulae for Bi_2_SmSbO_7_ and ZnBiYO_4_ could be determined. It can be seen from Figure 8 that various elemental peaks with specific binding energies were obtained. In Figure 8, the O1s peak of the O element is located at 529.90 eV. The position of the Bi5d_3/2_ and Bi5d_5/2_ peaks were fitted (located at 28.45 eV and 26.05 eV). The position of the Sm3d_5/2_ peak was at 1082.99 eV and the Sb4d_3/2_ peak was located at 35.20 eV. The position of Zn2p_1/2_ or Zn2p_3/2_ peaks were at 1041.72 eV and 1021.45 eV. The position of the Y3p_3/2_ peak of the Y element was at 301.05 eV. In short, Figure 7 and Figure 8 showed the existence of zinc (Zn_2p_), samarium (Sm_3d_), antimony (Sb_4d_), bismuth (Bi_4f_ and Bi_5d_), Yttrium (Y_3p_) and oxygen (O_1s_) within the synthetical catalysts. The results of the superficial element characterization showed that the medial atomistic percentage of Bi:Sm:Sb:Zn:Y:O was equivalent to 940:262:285:384:402:7727. The atomistic proportion of Bi:Sm:Sb or Zn:Bi:Y in the catalyst of BZHP was equivalent to 2.09:1.00:1.09 and 1.00:1.02:1.05, respectively. The reason for the high oxygen value might be owing to the large amount of O which was sorbed on the superficies of Bi_2_SmSbO_7_/ZnBiYO_4_. Obviously, it showed that there were no other phases in the XPS peak of Bi_2_SmSbO_7_/ZnBiYO_4_ because neither shoulder nor expansion was observed.

Figure 9 shows the SEM image of BZHP. Figure 10 shows the EDS elemental mapping of BZHP. Figure 11 shows the EDS spectra of BZHP. The results of Figure 9 and Figure 10 showed that the large dodecahedron structure belonged to Bi_2_SmSbO_7_, and the uniformly dispersed spherical flocculent small particles belonged to ZnBiYO_4_. Small particles of ZnBiYO_4_ were closely surrounded and loaded on the surface of large particles of Bi_2_SmSbO_7_. As could be seen from Figure 9 and Figure 10, the particles of Bi_2_SmSbO_7_ were surrounded by the smaller particles of ZnBiYO_4_ and all these particles were tightly bound together, which was a strong proof of the successful preparation of BZHP. Bi_2_SmSbO_7_ possessed a rhombic dodecahedron-like morphology. It was common sense that the distinct surface energy of crystallite facets controlled the structural growth of the photocatalyst. Previously, researchers have discovered that the Ag_3_PO_4_ (110) surface possessed a higher superficial energy when compared with the (111) surface, therefore, the crystal structure aggregated along the (110) direction, resulting in the formation of a rhombic dodecahedron-like morphology of silver phosphate [92,93]. It could be concluded that the reason why Bi_2_SmSbO_7_ possessed a diamond-shaped dodecahedron morphology might be illuminated in the above analysis. ZnBiYO_4_ possessed a regular spherical morphology and a uniform particle distribution (Figure 9). The particle size of Bi_2_SmSbO_7_ was approximately 2600 nm, while the particle size of ZnBiYO_4_ was approximately 1000 nm.

The results of the SEM-EDS analysis are shown in Figure 9, Figure 10 and Figure 11, and other impurities were not found in the BZHP compound. Similarly, the unmingled phase of Bi_2_SmSbO_7_ had a good agreement with the results of the XRD analysis, which is represented in Figure 1a. From Figure 10 and Figure 11, it can be concluded that BZHP contained bismuth element, samarium element, antimony element, zinc element, yttrium element and oxygen element. The previous results had a good agreement with the XPS results of BZHP. According to the EDS spectrum of BZHP (Figure 11), the atomic ratio of Bi:Sm:Sb:Zn:Y:O was 1213:414:418:405:389:7161, which was also consistent with XPS results of BZHP. The atomic ratio of Bi_2_SmSbO_7_:ZnBiYO_4_ was close to 1000:973. Based on the above results, we could conclude that BZHP owned high purity under the preparation conditions which were used in this work.

### 2.4. Photocatalytic Activity

Figure 12 shows the concentration variance bights of DO during PD of DO with BZH as catalyst or with Bi_2_SmSbO_7_, ZnBiYO_4_ and N-dT as catalyst under VLI. It could be found from Figure 12 that the concentration of DO within dye wastewater gradually decreased with increasing visible light irradiation time when Bi_2_SmSbO_7_/ZnBiYO_4_ heterojunction or Bi_2_SmSbO_7_ or ZnBiYO_4_ or N-dT was utilized as a catalyst for degrading DO. In all the contrast experiments, the VLI time was set to be 160 min. The results which were obtained from Figure 12 showed that after VLI-160M, with BZH as the catalyst, the RR value of DO in dye wastewater achieved 99.10%, the reactive rate was equivalent to 3.097 × 10^−9^ mol·L^−1^·s^−1^, and the photon efficiency (PE) was equivalent to 0.0651%. With Bi_2_SmSbO_7_ as the photocatalyst or with ZnBiYO_4_ as the photocatalyst, the RR decreased. When Bi_2_SmSbO_7_, ZnBiYO_4_ or N-dT were utilized as the photocatalyst, the RR of DO achieved 82.57%, 78.13% or 32.83%, the rate of reaction was equivalent to 2.58 × 10^−9^ mol·L^−1^·s^−1^, 2.44 × 10^−9^ mol·L^−1^·s^−1^ or 1.03 × 10^−9^ mol·L^−1^·s^−1^, and the PE was equivalent to 0.0542%, 0.0513% or 0.0216%, respectively. Therefore, it was obvious that the photodegradation efficiency of DO was the highest when using BZHP. By calculating the contrast experimental results, the RR of DO by using BZH was 1.200 or 1.268 or 3.019 times higher than that with Bi_2_SmSbO_7_ as the catalyst, ZnBiYO_4_ as the catalyst or with N-dT as the catalyst. Using Bi_2_SmSbO_7_ as the photocatalyst or BZHP as the photocatalyst, respectively, the concentration of nulvalent Sb or Sb^5+^ in the aqueous solution before photocatalytic degradation of direct orange was zero. After VLI-160M for PD of DO, the content of nulvalent Sb or Sb^5+^ in the aqueous solution was also zero. Regarding the specific surface area, the specific surface area of Bi_2_SmSbO_7_ was 4.15 m^2^/g, the specific surface area of ZnBiYO_4_ was 4.06 m^2^/g, and the specific surface area of BZHP was 4.12 m^2^/g. All of our photocatalytic reactions (Figure 12) were realized by the photocatalysts which were prepared by the hydrothermal synthesis method; thus, it would not cause a difference in photocatalytic activity.

Figure 13 shows the concentration changing curved line of total organic carbon (TOC) during PD of DO in dye wastewater with BZH or with Bi_2_SmSbO_7_ or with ZnBiYO_4_ or with N-dT as catalyst under VLI. The concentration of DO gradually decreased with increasing VLI time. As could be found from Figure 13, the RR of TOC within dye wastewater reached 96.21%, 73.54%, 68.71% and 25.78%, respectively, after VLI-160M when BZHP, Bi_2_SmSbO_7_, ZnBiYO_4_ and N-dT were used for degrading DO. In summary, based on all the above results, it was easy to conclude that the RR of TOC during removing DO when using BZHP was higher than that when Bi_2_SmSbO_7_, ZnBiYO_4_ or N-dT were used, which meant that BZHP owned the maximal mineralization percentage ratio compared with the other three photocatalysts.

Figure 14 presents the concentration variation curves of DO during PD with Bi_2_SmSbO_7_/ZnBiYO_4_ heterojunction as the photocatalyst under VLI for three cycle degradation (TCD) tests. Figure 14 shows that the RR of DO reached 98.03%, 96.73% or 95.43%, respectively, after VLI-160M with Bi_2_SmSbO_7_/ZnBiYO_4_ heterojunction as catalyst by finishing 3 cycle experiments for removing DO. Figure 15 reveals the concentration changing curved line of TOC during PD of DO with BZH as the photocatalyst under VLI for TCD tests. We could observe from Figure 15 that the RR of TOC was 94.98%, 93.51% or 92.19%, respectively, after VLI-160M with BZH as the photocatalyst. The experimental results, which were obtained from Figure 14 and Figure 15, showed that the BZHP had high stability.

Figure 16 exhibits the second-order kinetic curves for the PD of DO with BZH, Bi_2_SmSbO_7_, ZnBiYO_4_ or N-dT as catalysts under VLI. According to Figure 16, the dynamic constant k, which was obtained from the kinetic plot toward DO concentration and VLI time with BZH, Bi_2_SmSbO_7_, ZnBiYO_4_ or N-dT as catalyst, reached 2.167 or 0.495 or 0.395 or 0.089 min^−1^, respectively. The dynamic constant k, which derived from the kinetic plot toward TOC concentration, was 0.047 or 0.010 or 0.009 or 0.002 min^−1^ with BZH, Bi_2_SmSbO_7_, ZnBiYO_4_ or N-dT as photocatalysts. The fact that the merical number of *K_TOC_* for removing DO was lower than the merical number of *K_C_* for removing DO, even though they were using the same catalyst, indicated that the photodegradation intermediate products probably appeared during the PD of DO under VLI. At the same time, the degradation of DO by BZHP showed higher mineralization efficiency compared with the other three photocatalysts.

Figure 17 displays the observed second-order kinetic plots for the PD of DO with BZH as the photocatalyst under VLI for TCD tests. According to the results in Figure 17, the dynamic constant k, which was obtained from the kinetic plot towards the DO concentration and VLI time with BZH as the photocatalyst for TCD tests, was equivalent to 1.415 or 0.849 or 0.558 min^−1^. Figure 18 shows the achieved second-order dynamic curves for TOC during the PD of DO with BZH as catalyst under VLI for TCD tests. It could be found from Figure 18 that the kinetic constant k, which came from a dynamic curve towards the TOC concentration and VLI time with BZH as the photocatalyst for TCD tests, achieved 0.031 min^−1^ or 0.022 min^−1^ or 0.014 min^−1^. The results of Figure 16, Figure 17 and Figure 18 exhibited that the PD of DO with BZH as the photocatalyst under VLI coincided to the second-order reaction kinetics. 

A conclusion could be summarized from Figure 17 and Figure 18 that the RR of DO decreased by 3.67% with BZH as the photocatalyst under VLI after TCD tests and the RR of TOC decreased by 4.02%. In the above three cycle experiments, there was no significant difference in degradation efficiency, and the photocatalyst structure of BZHP was stable. 

Figure 19 exhibits the relation curves among ethylenediamine tetraacetic acid (EDTA), isopropanol (IPA) or benzoquinone (BQ) and RR of DO with BZH as the catalyst under VLI. At the beginning of the photo-catalysis experiment, different free radical scavengers were added to the DO solution to capture the active species during the degradation process of DO. Isopropanol (IPA) that we used to capture hydroxyl radicals (^•^OH), benzoquinone (BQ) that we utilized to capture superoxide anions (^•^O_2_^−^), and ethylenediaminetetraacetic acid (EDTA) that we used to capture holes (h^+^). The starting IPA concentration, BQ concentration or EDTA concentration was equivalent to 0.15 mmol L^−1^, and the added amount of IPA or BQ or EDTA was equivalent to 1 mL. Based on Figure 19, when the IPA, BQ or EDTA was put into the DO solution, the RR of DO decreased by 67.13%, 49.87% or 27.80%, respectively, compared with the standard RR of DO. Therefore, the conclusion could be drawn that in the process of DO degradation, ^•^OH, h^+^ and ^•^O_2_^−^ were all active free radicals, and ^•^OH played a leading role when using BZH as the photocatalyst to degrade DO under VLI. By using the experiment of adding a capture agent, it was found that the hydroxyl radical possessed the maximum oxidizing removal capability for eliminating DO in dye wastewater compared with superoxide anion or holes. The oxidizing removal capability for degradation DO was as follows: hydroxyl radical > superoxide anion > holes.

The Nyquist impedance plot measurement was an important test that was always used for characterization of the migrating course of photoinduced electrons and photoinduced holes at the solid/electrolyte separating surface of the photocatalysts. The smaller arc radius meant that the transportation efficiency of the photocatalysts was high. Figure 20 shows the corresponding Nyquist impedance plots of the prepared BZHP or Bi_2_SmSbO_7_ photocatalyst or ZnBiYO_4_ photocatalyst. It was distinct, according to Figure 20, that the diameter of the arc radius was in the order ZnBiYO_4_ > Bi_2_SmSbO_7_ > BZHP, as the above results indicated that BZHP exhibited a more efficient separation of photogenerated electron and photogenerated hole and faster interfacial charge migration ability. 

### 2.5. Probable Degradation Mechanism Analysis

The probable PD mechanism of DO with BZH as photocatalyst under VLI is exhibited in Figure 21. The potentials of the conductance band (CB) or valence bond band (VB) for quasi-conductors were estimated using the following Formulas (3) and (4) [94]:(3)$$E_{CB}=X-E^{e}-0.5E_{g}$$
(4)$$E_{VB}=E_{CB}+E_{g}$$

In the above two equations, *E_g_* was the bandgap of the quasi-conductor, *X* was the electronegativity of the quasi-conductor and *E^e^* was the free electronic energy on the hydrogen scale (*E^e^* = 4.5 eV). The VB electric potential or the CB electric potential for Bi_2_SmSbO_7_ (determined by Formulas (3) and (4)) was equivalent to 2.539 eV or 0.123 eV, respectively. In addition, for ZnBiYO_4_, the VB electric potential and the CB electric potential were calculated to be 1.271 eV and −0.682 eV, respectively. It could be found that both Bi_2_SmSbO_7_ and ZnBiYO_4_ could assimilate seeable light and constitutionally generated electron–hole pairs when the BZHP was irradiated by VLI. Since the redox potential position of the CB of ZnBiYO_4_ (−0.682 eV) was more negative than that of Bi_2_SmSbO_7_ (0.123 eV), the photoinduced electrons on the CB of ZnBiYO_4_ could transform to the CB of Bi_2_SmSbO_7_. In addition, the redox potential position of the VB of Bi_2_SmSbO_7_ (2.539 eV) was more positive than that of ZnBiYO_4_ (1.271 eV), the photoinduced holes on the VB of Bi_2_SmSbO_7_ could transfer to the VB of ZnBiYO_4_. Hence, using BZHP, which consists of Bi_2_SmSbO_7_ and ZnBiYO_4_, would obviously diminish the reunion rate of photo-induced electrons and photo-induced holes. Moreover, the inner resistance would also decrease, and the lifespan of photoinduced electrons, photo-induced holes and the interfacial charge transfer would be enhanced [95]. As a result, more ^•^OH or ^•^O_2_^−^ (oxidative radicals) could be manufactured, helping to raise the removal efficiency of DO. In Figure 21, the CB potential of ZnBiYO_4_ was −0.682 eV and the potential of O_2_/^•^O_2_^−^ was −0.33 V, and more subtractive potential meant that the electrons within the CB of ZnBiYO_4_ could absorb oxygen to produce ^•^O_2_^−^ which could degrade DO. The value of the VB electric potential of Bi_2_SmSbO_7_ was (2.539 eV) larger than that of OH^−^/^•^OH (2.38 V), revealing that the holes in the VB of Bi_2_SmSbO_7_ could oxidize H_2_O or OH^−^ into ^•^OH for degrading DO, which was shown as path 2. Lastly, as shown in path 3, the photoinduced holes in the VB of Bi_2_SmSbO_7_ or ZnBiYO_4_ could straightly oxidize and remove DO owing to its strong oxidation capability. To sum up, the high efficiency of electron–hole separation was the reason that BZHP could promote DO degradation.

For the purpose of studying the degradation mechanism of DO, the intermediate products were also detected using the LC-MS method during the degradation process of DO. The intermediate products which were obtained during the PD of DO were identified as phenyldiazene (*m*/*z* = 106), naphtahlene (*m*/*z* = 127), aniline (*m*/*z* = 93), hydroquinone (*m*/*z* = 112), 1,2,6-trihydroxy-3-naphthalene sulfonate (*m*/*z* = 257), phenol (*m*/*z* = 94), oxalid acid (*m*/*z* = 90), C_11_H_10_O_5_N_2_S (*m*/*z* = 282), C_11_H_9_O_5_NS (*m*/*z* = 266) and 8-aminonaphthol (*m*/*z* = 152). Based on the detected intermediates, we could extrapolate the degradation pathway of DO, as shown in Figure 22. It could be found from Figure 22 that oxidation reaction and hydroxylation reaction were realized during PD process of DO. Ultimately, DO was converted into small molecular organic compounds and finally united with other organic active groups to convert into carbon dioxide and water.

## 3. Experimental Section

### 3.1. Materials and Reagents

The analytical grades were ethylenediaminetetraacetic acid (EDTA, 99.5%), isopropyl alcohol (IPA, purity ≧ 99.7%) and *P*-benzoquinone (BQ, purity ≧ 98.0%). The purchased anhydrous ethanol (purity ≧ 99.5%) conformed to the specifications of the American Chemical Society. The gas chromatography grade was DO (chemical formula: C_33_H_22_N_6_Na_2_O_9_S_2_, purity ≧ 98%). In this work, ultra-pure water (18.25 MU cm) was utilized.

### 3.2. Preparation Method of Bi_2_SmSbO_7_

The new photocatalyst Bi_2_SmSbO_7_ was synthesized via a high-temperature solid-phase sintering method at a temperature of 1090 °C. High purity Bi_2_O_3_ (99.99%), Sm_2_O_3_ (99.99%) and Sb_2_O_5_ (99.99%) were used for raw materials. Because all the raw materials were of high purity, it was unnecessary to do the further purification test. Due to the high volatility of Bi_2_O_3_ at high temperatures, we finally decided to increase the amount of Bi_2_O_3_ to 120% after 5 experiments. Before the experiment, all the above powders (*n*(Bi_2_O_3_):*n*(Sm_2_O_3_):*n*(Sb_2_O_5_) = 2.4:1:1) were dried for 4 h at 200 °C. The Bi_2_SmSbO_7_ was prepared by mixing the precursors stoichiometrically, then pressing them into small columns and putting them in an alumina crucible. After calcination in an electric furnace for 2 h at 400 °C, the raw materials and small columns were taken out. We ground the mixture and then put them in the electric stove. Finally, it was calcined separately in an electric furnace at 1090 °C for 35 h. 

The 0.30 mol/L Bi (NO_3_)_3_·5H_2_O, 0.15 mol/L Sm(NO_3_)_3_·6H_2_O and 0.15 mol/L SbCl_5_ were blended and kept stirring for 20 h. The above solution was transferred to an autoclave lined with polytetrafluoroethylene and heated at 200 °C for 15 h. Afterwards, the achieved powder was calcined in a tubular stove at 8 °C/min under N_2_ for 10 h at 800 °C. Finally, Bi_2_SmSbO_7_ powder was also obtained by the hydrothermal synthesis method.

### 3.3. Preparation Method of ZnBiYO_4_

Preparation of the ZnBiYO_4_ catalyst was realized by high-temperature solid-phase sintering method. ZnO, Bi_2_O_3_ and Y_2_O_3_ were all raw materials with a purity as high as 99.99%. Due to the high volatility of Bi_2_O_3_ at high temperatures, we finally decided to increase the amount of Bi_2_O_3_ to 120% after 5 experiments. In order to decrease the particle size of the fully-mixed materials (*n*(ZnO): *n*(Bi_2_O_3_): *n*(Y_2_O_3_) = 2.4:1:1), a ball mill method was used for making the final particle size to 1–2 µm. Before the synthesis of the target products, all the powder compounds were dried at 200 °C for 4 h. These powders were pressed into discs and put into an alumina crucible in an electric stove (KSL1700X, Hefei Kejing Materials Technology Co., Ltd., Hefei, China) and heated at 750 °C for 6 h. The powder was heated again in the same electric furnace at 1000 °C for 35 h after the pressing and crushing procedures. Lastly, after complete grinding, pure ZnBiYO_4_ catalyst was obtained.

The 0.15 mol/L Zn(NO_3_)_2_·6H_2_O, 0.15 mol/L Bi (NO_3_)_3_·5H_2_O and 0.15 mol/L Y(NO_3_)_3_·6H_2_O were mixed and kept stirring for 20 h. The above solution was transferred to an autoclave lined with polytetrafluoroethylene and heated for 15 h at 200 °C. Then, the resultant powder was calcined in a tubular stove at a rate of 8 °C per min under N_2_ protection at 780 °C for 10 h. Finally, ZnBiYO_4_ powder was also obtained by hydrothermal synthesis method.

### 3.4. Synthesis of N-Doped TiO_2_

The nitrogen-doped titanium dioxide (NT) catalyst used tetrabutyl titanate as the precursor, ethanol as the solvent, and was prepared by the sol-gel method. The operation steps were as follows: the first step was to mix 17 mL of tetrabutyl titanate with 40 mL of absolute ethanol to form solution A; 40 mL of absolute ethanol, 10 mL of glacial acetic acid and 5 mL of double distilled water were mixed to make solution B. Under stirring conditions, solution A was mixed dropwise into the solution to form a transparent colloidal suspension (TCS). In the second step, under magnetic stirring conditions, ammonia water with an N/Ti ratio of 8 mol% was mixed with the obtained TCS for 1 h. In the third step, a xerogel was formed after aging for two days. The dry gel was ground into powder and calcined for 2 h at 500 °C. In the final step, we ground the powder in the agate mortar and sieved it through a vibrating screen to acquire NT powder.

### 3.5. Synthesis of Bi_2_SmSbO_7_/ZnBiYO_4_ Heterojunction Photocatalyst

The maximum calcination temperature of ZnBiYO_4_ which was prepared by the solid-state sintering method was 1000 °C and the heat retaining time was 35 h. The maximum calcination temperature of Bi_2_SmSbO_7_, which was prepared by the solid-state sintering method was 1090 °C and the heat retaining time, was 35 h. The highest calcination temperature of ZnBiYO_4_ which was prepared by the hydrothermal synthesis method was 780 °C, and the heat retaining time was 10 h. The maximum calcination temperature of Bi_2_SmSbO_7_, which was prepared by hydrothermal synthesis method, was 800 °C and the heat retaining time was 10 h. On the one hand, the higher the maximum calcination temperature was, the greater the power energy consumption was, which would reduce and consume the service life of the furnace instrument. On the other hand, the longer heat retaining time and the higher maximum sintering temperature would cause the larger particle size of ZnBiYO_4_ or Bi_2_SmSbO_7_. As a result, the specific surface area of ZnBiYO_4_ or Bi_2_SmSbO_7_ would be reduced and the photocatalytic activity of ZnBiYO_4_ or Bi_2_SmSbO_7_ would be correspondingly decreased. In order to improve the photocatalytic activity, reduce energy consumption and improve the instrument life of high-temperature calciner, we used the hydrothermal synthesis method to prepare ZnBiYO_4_ and Bi_2_SmSbO_7_ in the process of preparing heterojunction.

First, solution Bi (NO_3_)_3_·5H_2_O (0.30 mol/L), solution Sm(NO_3_)_3_·6H_2_O (0.15 mol/L) and solution SbCl_5_ (0.15 mol/L) were mixed and kept stirring for 20 h. The above solution was transferred to an autoclave lined with polytetrafluoroethylene and heated at 200 °C for 15 h. Afterwards, the achieved powder was calcined at 800℃ for 10 h in a tubular stove at a rate of 8 °C per minute under N_2_ protecting. Bi_2_SmSbO_7_ powder was obtained by hydrothermal synthesis method. Secondly, solution Zn(NO_3_)_2_·6H_2_O (0.15 mol/L), solution Bi (NO_3_)_3_·5H_2_O (0.15 mol/L) and solution Y(NO_3_)_3_·6H_2_O (0.15 mol/L) were mixed and kept stirring for 20 h. The above solution was transferred into an autoclave lined with polytetrafluoroethylene and heated at 200 °C for 15 h. Then, the achieved powder was calcined at 780 °C for 10 h in a tubular stove at a rate of 8 °C per minute under N_2_ protecting. ZnBiYO_4_ powder was obtained by hydrothermal synthesis method. The individual photocatalysts, such as ZnBiYO_4_ or Bi_2_SmSbO_7_, were prepared by hydrothermal synthesis method.

In this text, a new catalyst BZHP was synthesized by solvothermal method. BZHP was prepared by mixing 890 mg Bi_2_SmSbO_7_ and 30 wt.% (610 mg) ZnBiYO_4_ in 300 mL of octanol (C_8_H_18_O) and the above mixture was dispersed in an ultrasonic bath for 1 h. Then, under vigorous stirring conditions, the mixture was heated to reflux at 140 °C for 2 h to improve the adhesion of ZnBiYO_4_ on the surface of Bi_2_SmSbO_7_ nanoparticles and BZHP was formed. First, the catalyst was naturally cooled to room temperature, then the products were collected by centrifugation method and washed with a hexane / ethanol mixture. After the powder was purified, the powder was dried in a 60 °C vacuum oven for 6 h and laid in a desiccator for later use. Finally, BZHP was prepared successfully.

### 3.6. Characterizations

The structure of the samples was analyzed using a powder XRD test (Cu Kɑ radiation, λ = 1.54184 Å, preset time of 0.3 s step^−1^, step length of 0.02°). A scanning electron microscope (SEM) was used to characterize the morphology and microstructure of the prepared samples and the elementary composition, which was derived from above prepared samples, was obtained by energy dispersive spectroscopy (EDS). The diffuse reflectance spectra of the above prepared samples were obtained by UV-Vis spectrophotometer (UV-Vis DRS, UV-3600). Valence analysis and chemical composition of surface for the catalysts were realized by X-ray photoelectron spectrograph (XPS) with an Al-kα X-ray source. 

### 3.7. Photoelectrochemical Experiments

The electrochemical impedance spectroscopy experiment was performed by the CHI660D electrochemical station with standard 3 electrodes. In this system, the working electrode, counter electrode and reference electrode are prepared catalyst, platinum plate and commercial Ag/AgCl electrode, respectively. An aqueous solution of Na_2_SO_4_ (0.5 mol/L) was used as the electrolyte, and a 500 W xenon lamp with an ultraviolet cut-off filter was used as the visible light lamp for photochemical measurement. The working electrode was prepared by the following method: Dissolved 0.03 g of the sample and 0.01 g of chitosan in 0.45 mL of dimethylformamide, and to form a uniform suspension after ultrasonic treatment for 1 h. Subsequently, they were dropped on indium tin oxide (ITO) conductive glass with a size of 1 cm × 2 cm. Finally, we dried the working electrode at 80 °C, which lasted for 10 min.

### 3.8. Experimental Setup and Procedure

The temperature of the experimental reaction system was 20 °C (reactive vessel, XPA-7, Xujiang Electromechanical Plant, Nanjing, China), which was regulated by circulating cooling water. The simulated daylight illumination consists of a 500 W xenon lamp and a 420 nm cut-off filter. There were 12 same quartz tubes (40 mL). The dosage of Bi_2_SmSbO_7_ or ZnBiYO_4_ or BZHP was equivalent to 0.75 g/L. Moreover, the concentration of DO was equivalent to 0.03 mmol/L. The DO concentration was the residuary concentration of actual dye wastewater after biodegradation, and the content of DO was equivalent to 1.2 mmol/L. During the reaction, 3 mL of suspension was withdrawn termly. Subsequently, the filtration was realized for removing the catalyst. Ultimately, the residuary concentration of DO in solution was defined by the UV-Vis spectrophotometer (Shimadzu, UV-2450, Shimadzu Corporation Co., Ltd., Chengdu, China). The absorption wavelength (detecting wavelength) of DO was 665 nm. The absorbance standard curve of DO at different concentrations was accomplished under ultraviolet light irradiation in the range of 220 nm–320 nm with an ultraviolet-visible spectrophotometer. The relationship between the concentration of DO and the absorbance value at 665 nm should be calculated. The absorbance of DO in the solution was measured at the absorption wavelength of 665 nm, the calibration curve of DO was drawn and a linear regression method was used for the quantification of DO. Before VLI, the suspension containing photocatalyst and DO was magnetically stirred in the dark for 45 min to establish adsorption/desorption equilibrium among photocatalyst, DO and O_2_. During the VLI, the suspension was agitated at 500 rpm. 

Experimental data of mineralization of DO in reaction solution were meteraged. In order to examine the TOC concentration during the process of PD of DO, potassium phthalate (KHC_8_H_4_O_4_) or anhydrous sodium carbonate was used as a standardized agent. Potassium phthalate standard solutions with carbon concentration (0–100 mg/L) were prepared for calibration purpose. Each time, we used 6 samples (45 mL) to measure TOC concentration.

Liquid chromatography-mass spectrometry was used to identify and measure DO, and its intermediate degradation products. Then, the 20 μL solution which was acquired after the photocatalytic reactivity was automatically injected. The flow rate was 0.2 mL/min, which was a mobile phase containing 60% methanol and 40% ultrapure water. Electrospray ionization interface (27 °C, 19.00 V), spray voltage of 5000 V, and constant sheath gas flow rate were MS conditions. Spectra were acquired in the *m*/*z* range from 50 to 600 in negative ion scan mode. 

In order to measure the photon intensity of incident light, the filter, which was 7 centimeters in length and 5 cm in width, was chosen to be irradiated by incident single-wavelength visible light of 420 nm. According to the formula of *υ* = *c*/*λ* and *hv*, which represented the energy of a photon, Avogadro constant *N_A_*, Planck constant *h*, photonic frequency *υ*, incident light wavelength *λ* and light velocity *c* were used to obtain the mole number of the total photons or the reactive photons which passed through the total area of above filter per unit time. The length between the xenon lamp and the light reactor was adjusted. As a result, the incident photon flux on the photoreactor was changed.

We estimated the photon efficiency according to the following Formula (5): (5)$$\varphi =\frac{R}{I_{0}}$$

In this formula, *ϕ* presented the photonic efficiency (%), *R* presented the degradation rate of DO (mol L^−1^ s^−1^), and *I_o_* presented the incident photon flux (Einstein L^−1^ s^−1^). The incident photon flux *I_o_* was measured by a radiometer under VLI. (*I_o_* = 4.76 × 10^−6^ Einstein L^−1^ s^−1^.)

## 4. Conclusions

For the first time, the Bi_2_SmSbO_7_ compound was successfully synthesized by two methods: the hydrothermal synthesis method and solid state method with high temperature. BZHP was proposed and synthesized with the solvothermal method for degrading DO in dye wastewater. The photophysical properties of the single phase Bi_2_SmSbO_7_ and BZH were investigated. In interpreting the results, the following conclusions could be easily obtained. Bi_2_SmSbO_7_ compound was a pure phase which crystallized in a pyrochlore structure that belonged to a cubic crystal system with the space group *Fd3m*. The lattice parameter *a* = 10.835(1) Å, and the BGE of Bi_2_SmSbO_7_ was 2.42 eV. BZHP was certified to be an efficient photocatalyst for remedying DO in the dye wastewater, after VLI-160M, the RR of DO was as high as 99.10%, and the RR of TOC was 96.21%. BZH showed the best performance in removing DO, the RR with BZH as the catalyst was equivalent to 1.200 or 1.268 or 3.019 times higher than the RR with Bi_2_SmSbO_7_ as the catalyst, ZnBiYO_4_ as the catalyst or with N-dT as the catalyst. Therefore, the study with BZH as the catalyst provided a new idea for the treatment of dye wastewater that contained DO and it could also promote the property improvement of the photocatalyst in the future.

## Figures and Tables

**Figure 1 materials-15-03986-f001:**
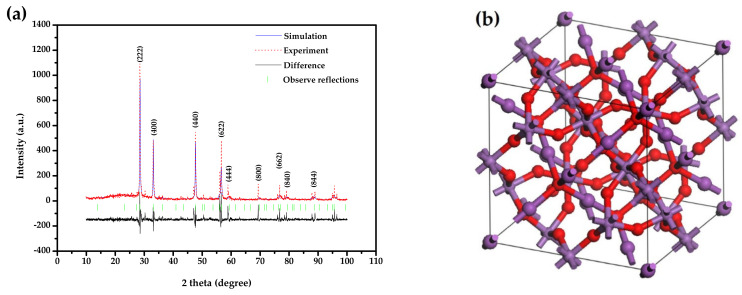
(**a**) XRD and corresponding Rietveld truing of Bi_2_SmSbO_7_ (red dotted line represents experimentative XRD datum of Bi_2_SmSbO_7_; blue solid line represents simulative XRD data of Bi_2_SmSbO_7_; black solid line is the disparity between experimentative XRD datum of Bi_2_SmSbO_7_ and analogous XRD datum of Bi_2_SmSbO_7_; green perpendicular is the observed reflective locality); (**b**) Atomy construction of Bi_2_SmSbO_7_. (Rubious atomy: O, dark purple atomy: Bi, light purpure atomy: Sm or Sb.).

**Figure 2 materials-15-03986-f002:**
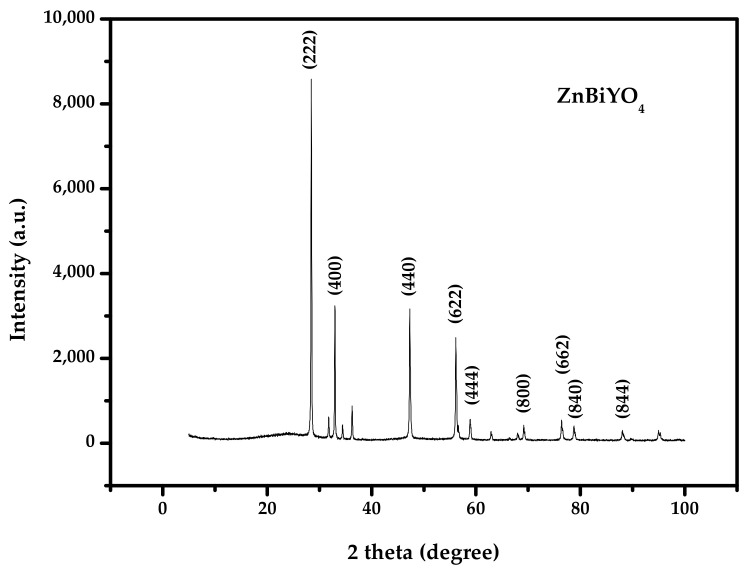
The XRD spectrum of ZnBiYO_4_.

**Figure 3 materials-15-03986-f003:**
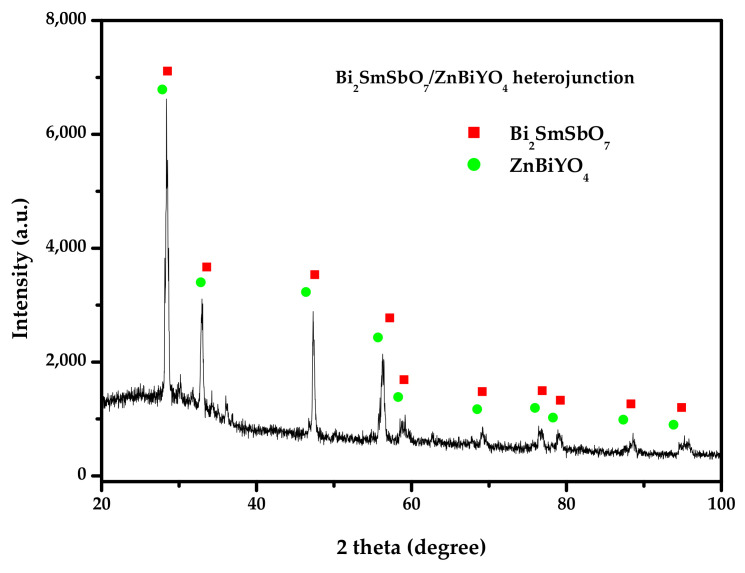
The X-ray diffraction spectrum of Bi_2_SmSbO_7_/ZnBiYO_4_ heterojunction.

**Figure 4 materials-15-03986-f004:**
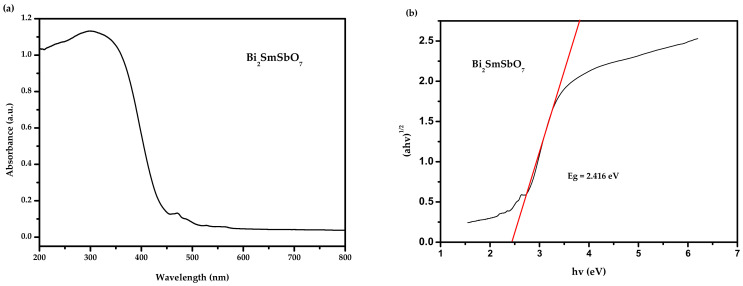
(**a**) The diffused reflection spectrum of Bi_2_SmSbO_7_; (**b**) Correlative diagram of (*αhν*)^1/2^ and *hν* for Bi_2_SmSbO_7_.

**Figure 5 materials-15-03986-f005:**
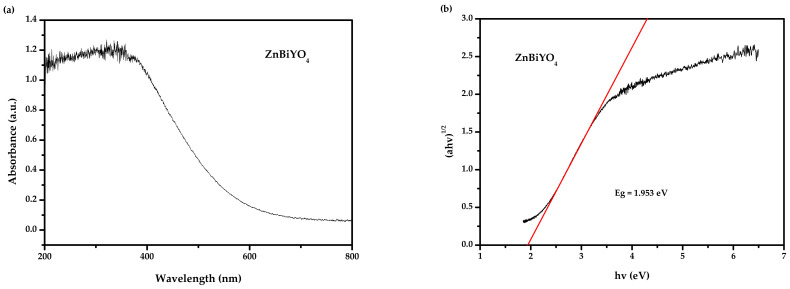
(**a**) The diffused reflection spectrum of ZnBiYO_4_; (**b**) Correlative diagram of (*αhν*)^1/2^ and *hν* for ZnBiYO_4_.

**Figure 6 materials-15-03986-f006:**
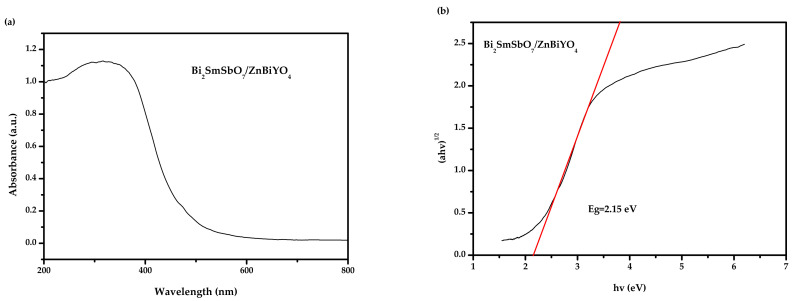
(**a**) The diffused reflection spectrum of Bi_2_SmSbO_7_/ZnBiYO_4_ heterojunction; (**b**) Correlative diagram of (*αhν*)^1/2^ and *hν* for Bi_2_SmSbO_7_/ZnBiYO_4_ heterojunction.

**Figure 7 materials-15-03986-f007:**
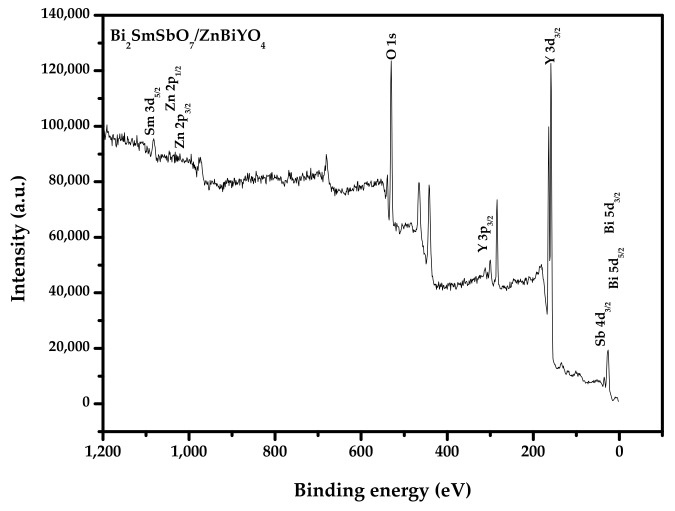
XPS survey spectrum of the Bi_2_SmSbO_7_/ZnBiYO_4_ heterojunction photocatalyst.

**Figure 8 materials-15-03986-f008:**
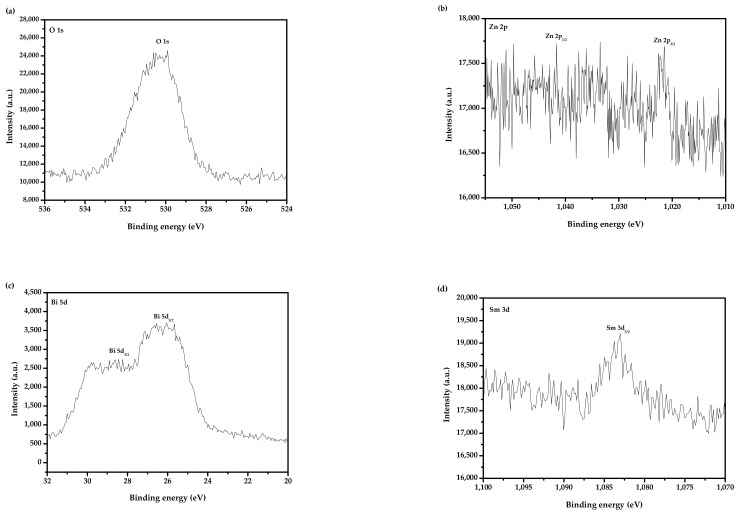
(**a**) XPS of O^2−^ which derived from BZHP; (**b**) XPS of Zn^2+^ which came from BZHP; (**c**) XPS of Bi^3+^ (Bi_5d_) which came from BZHP; (**d**) XPS of Sm^3+^ which came from BZHP; (**e**) XPS of Y^3+^ which came from BZHP; (**f**) XPS of Sb^5+^ which came from BZHP; (**g**) XPS of Bi^3+^ (Bi_4f_) which was obtained from BZHP.

**Figure 9 materials-15-03986-f009:**
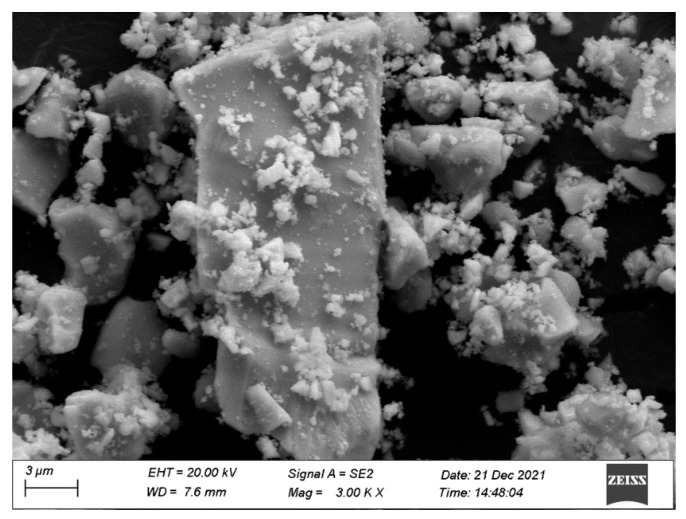
SEM photograph of Bi_2_SmSbO_7_/ZnBiYO_4_ heterojunction photocatalyst.

**Figure 10 materials-15-03986-f010:**
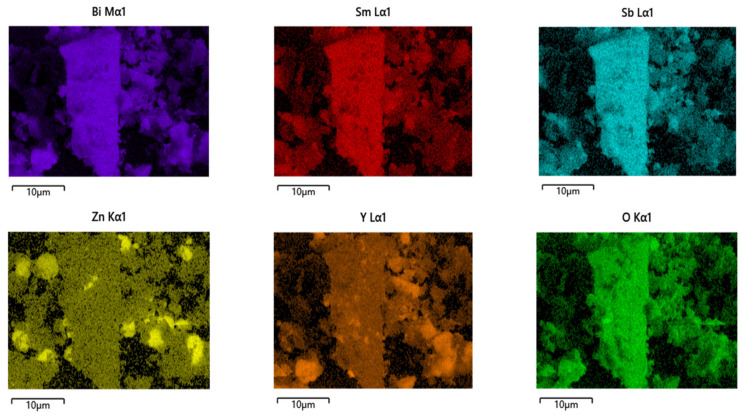
EDS elementary mapping of Bi_2_SmSbO_7_/ZnBiYO_4_ heterojunction catalyst (Bi, Sm, Sb, O from Bi_2_SmSbO_7_ and Zn, Bi, Y, O from ZnBiYO_4_).

**Figure 11 materials-15-03986-f011:**
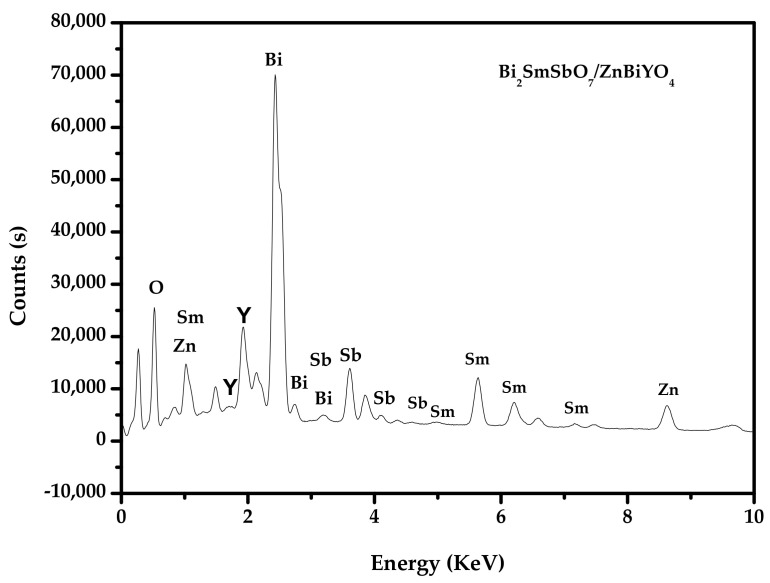
EDS spectrum of Bi_2_SmSbO_7_/ZnBiYO_4_ heterojunction photocatalyst.

**Figure 12 materials-15-03986-f012:**
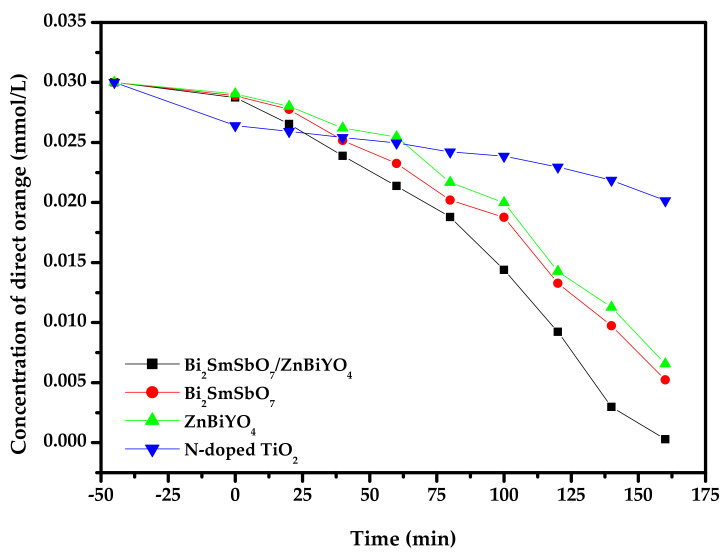
Concentration variation curves of DO during PD of DO with Bi_2_SmSbO_7_/ZnBiYO_4_ heterojunction as catalyst or with Bi_2_SmSbO_7_, ZnBiYO_4_, N-doped TiO_2_ as catalyst under VLI.

**Figure 13 materials-15-03986-f013:**
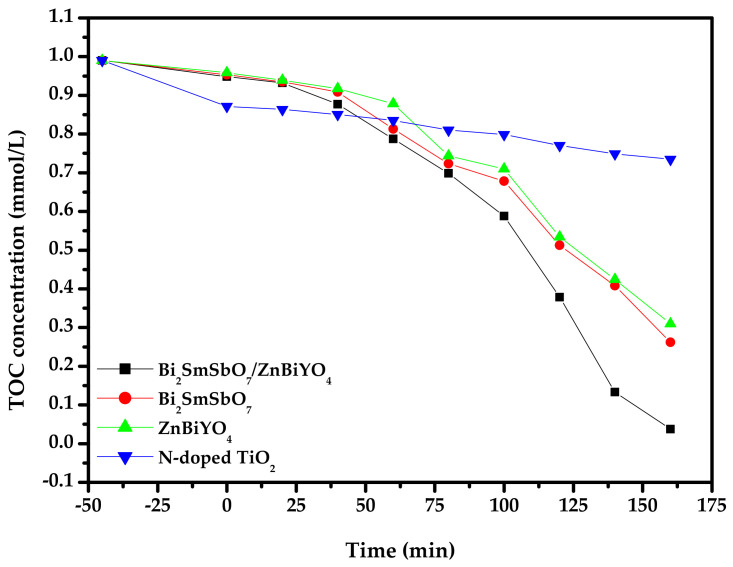
Concentration changing curved line of TOC during PD of DO in dye wastewater with Bi_2_SmSbO_7_/ZnBiYO_4_ heterojunction or with Bi_2_SmSbO_7_or with ZnBiYO_4_ or with N-doped TiO_2_ as catalyst under VLI.

**Figure 14 materials-15-03986-f014:**
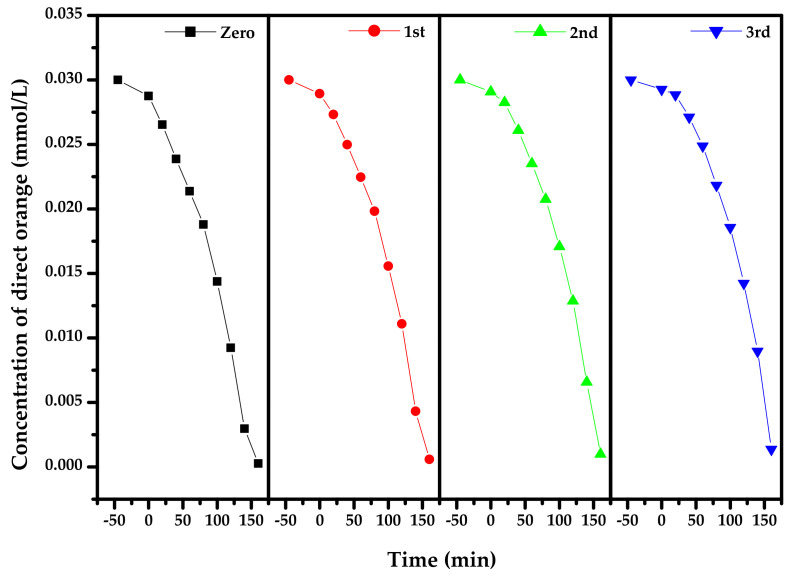
Concentration variation curves of DO during PD of DO in dye wastewater with Bi_2_SmSbO_7_/ZnBiYO_4_ heterojunction as photocatalyst under VLI for three cycle degradation tests.

**Figure 15 materials-15-03986-f015:**
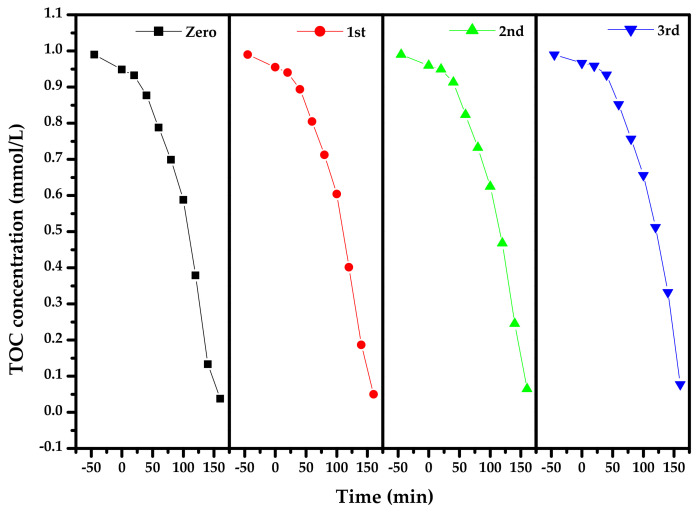
Concentration changing curved line of TOC during PD of DO in dye wastewater with Bi_2_SmSbO_7_/ZnBiYO_4_ heterojunction as photocatalyst under VLI for 3 cyclical degradation experiments.

**Figure 16 materials-15-03986-f016:**
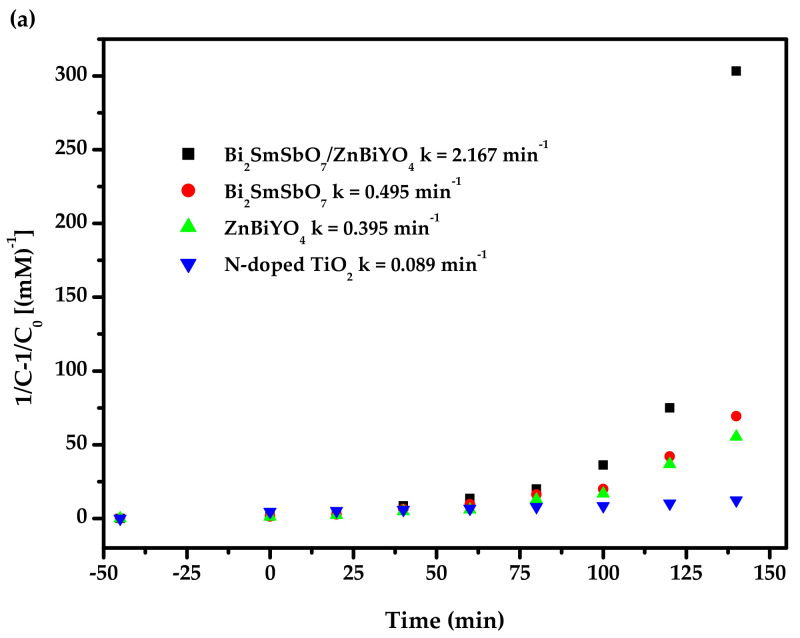
(**a**) Observed second-order dynamic curves for the PD of DO with Bi_2_SmSbO_7_/ZnBiYO_4_ heterojunction or with Bi_2_SmSbO_7_ or with ZnBiYO_4_ or with N-dT as catalyst under VLI; (**b**) Achieved second-order kinetic plots for TOC during PD of DO with BZH or with Bi_2_SmSbO_7_ or with ZnBiYO_4_ or with N-dT as catalyst under VLI.

**Figure 17 materials-15-03986-f017:**
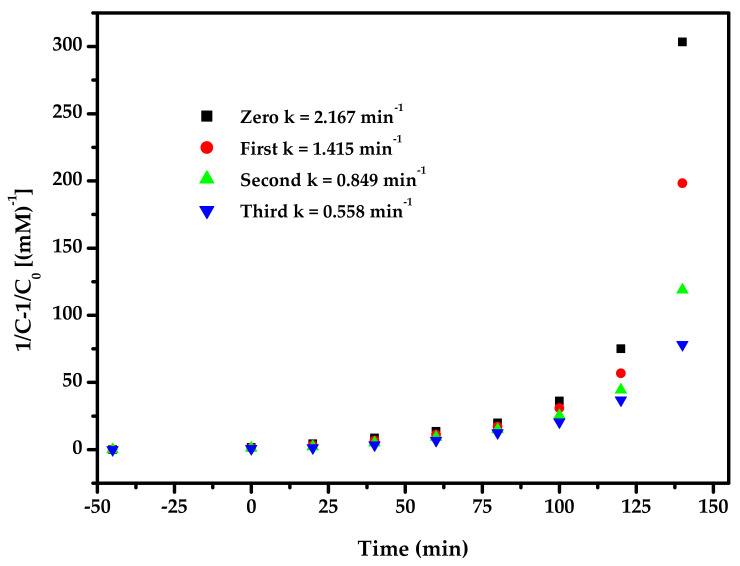
Achieved second-order dynamic curves for the PD of DO with Bi_2_SmSbO_7_/ZnBiYO_4_ heterojunction as the photocatalyst under VLI for three cycle degradation tests.

**Figure 18 materials-15-03986-f018:**
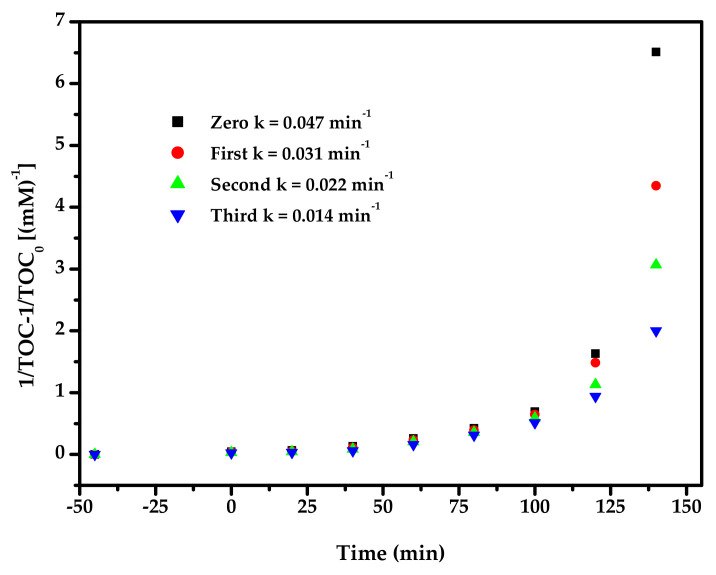
Observed second-order kinetic plots for TOC during the PD of DO with Bi_2_SmSbO_7_/ZnBiYO_4_ heterojunction as the photocatalyst under VLI for 3 cyclical degradation experiments.

**Figure 19 materials-15-03986-f019:**
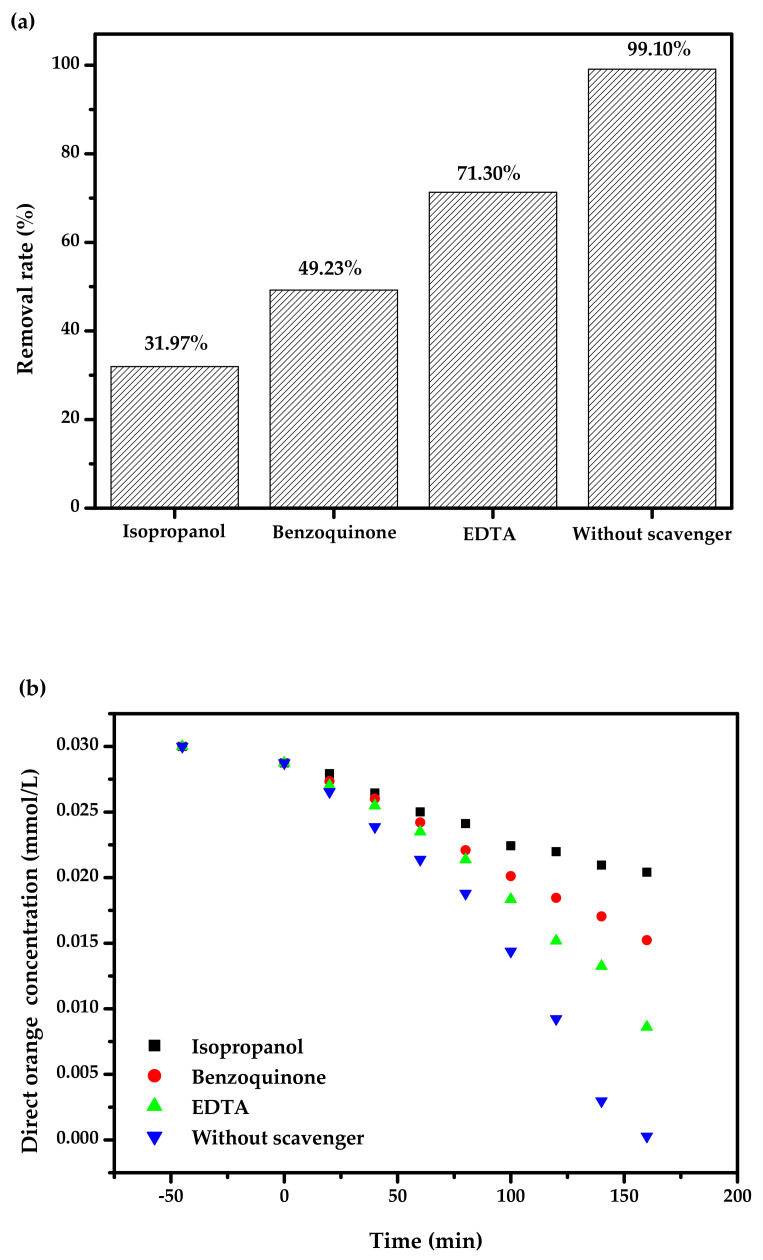
(**a**) RR of DO by using three trapping agents with BZH as the catalyst under VLI. (**b**) The relation curves among ethylenediamine tetraacetic acid (EDTA), isopropanol (IPA) or benzoquinone (BQ) and RR of DO with BZH as the catalyst under VLI.

**Figure 20 materials-15-03986-f020:**
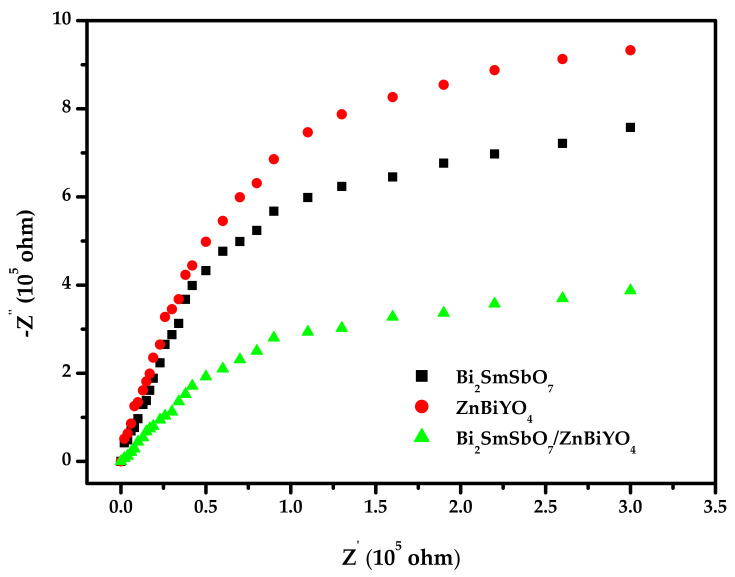
Nyquist impedance plots of Bi_2_SmSbO_7_/ZnBiYO_4_ heterojunction photocatalyst or Bi_2_SmSbO_7_ photocatalyst or ZnBiYO_4_ photocatalyst.

**Figure 21 materials-15-03986-f021:**
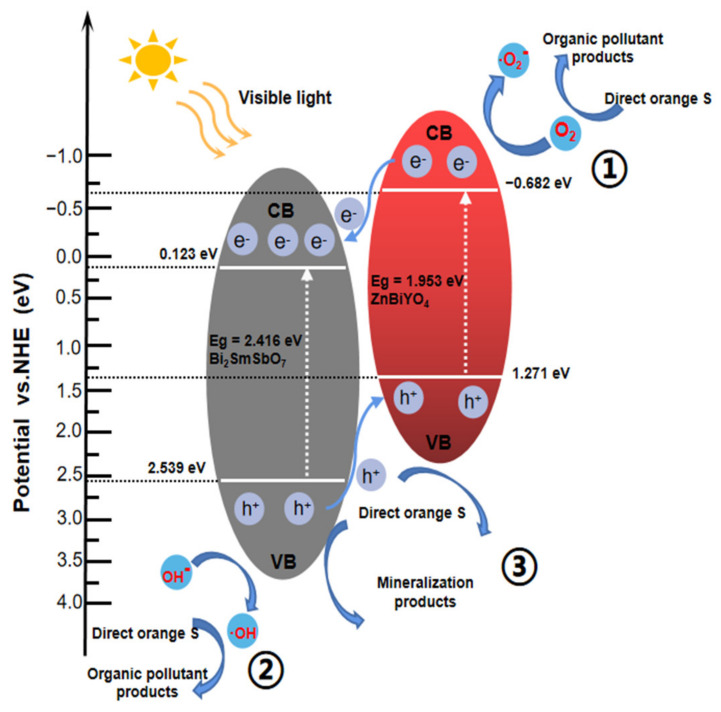
Possible PD mechanism of DO with Bi_2_SmSbO_7_/ZnBiYO_4_ heterojunction as photocatalyst under VLI.

**Figure 22 materials-15-03986-f022:**
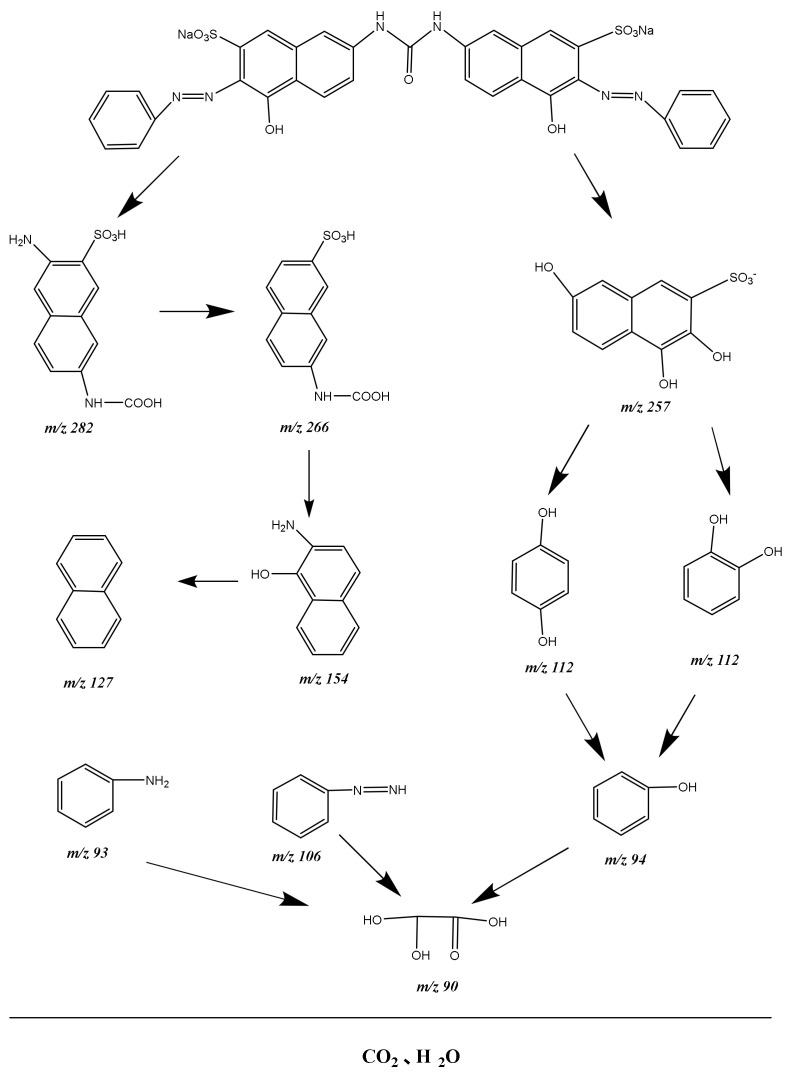
Suggested PD pathway scheme for DO under VLI with Bi_2_SmSbO_7_/ZnBiYO_4_ heterojunction as catalyst.

**Table 1 materials-15-03986-t001:** Crystallinenature parameters of Bi_2_SmSbO_7_.

Atomy	x	y	z	Occupation Factor
Bi	0	0	0	1
Sm	0.5	0.5	0.5	0.5
Sb	0.5	0.5	0.5	0.5
O(1)	−0.185	0.125	0.125	1
O(2)	0.125	0.125	0.125	1

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
