# Peer review of "Synthesis, Performance Measurement of Bi2SmSbO7/ZnBiYO4 Heterojunction Photocatalyst and Photocatalytic Degradation of Direct Orange within Dye Wastewater under Visible Light Irradiation"

_materials, 2022, doi:10.3390/ma15113986_

Round 1

Reviewer 1 Report

The authors report on the synthesis of Bi2SmSbO7/ZnBiYO4 composite photocatalysts and their application as visible light photocatalysts. The materials were characterized by several methods and their photocatalytic performance was evaluated on direct orange dye under visible light including scavenger experiments and intermediate product determination. Overall, the work is well-aimed and reports interesting data on the fabrication of efficient visible light heterojunction photocatalysts for dye degradation. However, the following points should be carefully considered and accordingly revised:

1) The abstract should be revised in order to become shorter and more concise describing briefly the purpose of the research, the principal results and major conclusions. In addition, the use of non-standard abbreviations should be avoided in the abstract. They should be better introduced in the manuscript text.

2) The use of many decimal points for the values of the kinetic constants, band gap etc. should be justified by the corresponding standard errors otherwise they should be reduced in the abstract and throughout the manuscript.

3) The XRD analysis and optical properties were reported only for Bi2SmSbO7 crystallizing in the pyrochlore structure. What about the ZnBiYO4 constituent? The corresponding structural data or appropriate references should be reported for this material, too. The bandgap of this oxide must be also reported/determined as it used in the construction of the proposed energy level diagram of the heterojuction structure in Figure 17.    

4) Another concern is that two different fabrication methods are reported for the synthesis of the two oxides, Bi2SmSbO7 and ZnBiYO4, that compose the heterojunction, namely the solid state reaction and the sol-gel/hydrothermal method. The latter materials seem to be used for the preparation of the composite structure. This point should be clarified and accordingly revised.

5) The authors compare the reaction constants derived for first order reaction kinetics (Figures 13 and 14). However, the corresponding plots deviate significantly form linearity, indicating that the reaction does not follow a first order law. This point should be accordingly revised.

6) There are several typos that should be corrected e.g. line 64: ZnO2 instead of ZnO, line 115: ZnBiTO4 instead of ZnBiYO4, line 127: single catalyst instead of single-phase. In addition, a polishing of the English would be required in order to improve the manuscript readability.

Reviewer 2 Report

The article "Synthesis, performance measurement of Bi2SmSbO7/ZnBiYO4 heterojunction photocatalyst and photocatalytic degradation of direct orange within dyewater under visible light irradiation" by Jingfei Luan et al. is devoted to the synthesis of a visible light efficient heterojunction photocatalyst Bi2SmSbO7/ZnBiYO4.

In the course of reviewing the presented results, I had a number of questions:

  1. First of all, I would like to draw attention to the fact that Sb is very toxic, the maximum permissible concentration in water is within µg/L. It would be good if its content in water was monitored before and after the process.
  2. From the introduction to the article it is not clear what justified the choice of ZnBiYO4 as the second component, please add.
  3. Please explain why the reference Bi2SmSbO7 and ZnBiYO4 photocatalysts were synthesized by the solid-phase method, while the Bi2SmSbO7/ZnBiYO4 heterostructure was synthesized by the sol-gel method. It seems to me that it is incorrect to compare from photocatalytic activity, since the difference in activity to a greater extent may be due to morphology, given the small difference (Figure 9). It is necessary to provide comparative data (SEM, BET surface area) to exclude this factor.
  4. It is necessary to present and analyze XRD spectra for Bi2SmSbO7/ZnBiYO4 and ZnBiYO4.
  5. It is necessary to submit UV-Vis diffuse reflectance spectra for Bi2SmSbO7/ZnBiYO4 and ZnBiYO4 and compare the band gap..
  6. Please show the high resolution spectrum for Bi 4f
  7. I do not quite understand the meaning of lines 223-225: “Figure 4 and Figure 5 223 revealed the presence of gallium (Ga2p), ytterbium (Yb4d), antimony (Sb3d and Sb4d), bismuth (Bi4f), iron (Fe2p), tantalum (Ta4f) and oxygen (O1s) within the prepared sample". Please explain.
  8. Please clarify how the incident photon flux was determined and what is the value

Round 2

Reviewer 1 Report

The authors have satisfactorily addressed my previous concerns and now the manuscript can be accepted in Materials.

Author Response

Dear reviewer

   We have revised our manuscript (materials-1560761) for decreasing the repetitive rate of this manuscript according to the comments. Please kindly find the red words in the revised manuscript. 

Best regards

Jingfei Luan

Reviewer 2 Report

I agree with the answers given and the changes made. I have one remark that needs to be corrected and given appropriate explanations in the text. In figure 6b, extrapolation should be made to zero on the y-axis, and not to -0.5 as you have. The band gap value is set incorrectly. Please clarify this
